# A Systematic Review on Carbon Dioxide (CO₂) Emission Measurement Methods under PRISMA Guidelines: Transportation Sustainability and Development Programs

Muhammad Zubair [1] 🅓, Shuyan Chen [1,2,*], Yongfeng Ma [1,2] 🅓 and Xiaojian Hu [1]

1 Jiangsu Key Laboratory of Urban ITS, School of Transportation, Southeast University, Nanjing 211189, China; 233217035@seu.edu.cn (M.Z.)
2 Jiangsu Province Collaborative Innovation Center of Modem Urban Traffic Technologies, Southeast University, Nanjing 211189, China
* Correspondence: chenshuyan@seu.edu.cn

**Abstract:** In the effort to urgently develop new and improved methodologies to compute and assess CO₂ emissions from transportation, results have been less than ideal; this article provides a review of the methodologies currently available. When it comes to the discharge of harmful gases into the air, transportation is the biggest offender. Methods are still being developed to calculate and analyze the transportation sector's carbon footprint, despite the fact that the need to limit the emission of gases that contribute to global warming has now become urgent. Previous studies have calculated the carbon footprint of transportation; however, there are some discrepancies in the terminology and methodologies utilized. The commonalities between CO₂ emission measurement techniques and assessment techniques are the primary subject of this review. This study helps to increase public awareness of environmental concerns and promotes the use of reliable methodologies for calculating transportation-related CO₂ emissions. It is hoped that choosing the optimal available method will contribute to a decrease in CO₂ emissions from transportation.

**Keywords:** vehicle emissions; CO₂ reduction method; greenhouse gases; sustainable development; implications and challenges





## 1. Introduction

To achieve national modernization and raise the standard of living for all citizens, it is necessary to engage in economic activities and development initiatives. However, the environmental issues that arise as a result of carrying out such economic operations and development initiatives are generally disregarded. Extensive damage to the natural world has resulted from hasty urbanization. This damage has a negative effect on people's health, the economy, and the environment. Pollution occurs when something new is introduced into the environment as a result of human activity and this new thing has the potential to harm or disrupt the natural ecosystem. Because of the dangers posed to human and environmental health, according to the Malayasian Environmental Quality Act (EQA), any release or removal of any substance, or any waste disposal method, that modifies the local environment, physically, chemically, or biologically, or that affects its radiation levels, can be classified as pollution. In Malaysia, the greenhouse gas (GHG) effect can be directly attributed to air pollution [1].

Any gas in the atmosphere that absorbs enough heat can be considered a greenhouse gas. The term received its name because greenhouses are where many plants are grown. The building's temperature is maintained via transparent panes that allow natural light to enter. After being warmed by the sun, the building's air might escape. This is a good analogy for the greenhouse effect that happens in the atmosphere. The greenhouse effect occurs when the GHG molecules of soak up the sun's heat and then radiate it back to the

earth's surface. The result is a general warming of the earth's atmosphere and surface. Thus, increased levels of atmospheric greenhouse gases will result in more radiation reflection, leading to elevated temperatures. A GHG effect, which helps keep the planet's average temperatures relatively constant, is, of course, a crucial phenomenon. Without it, the earth would cool to a chilly $-18$ degrees Celsius (compared to a comfortable $+14$ degrees Celsius). However, we now know that this GHG effect is not the same thing as global warming. Rather, the atmosphere is warming up because of high GHG concentrations. Carbon dioxide ($CO_2$), nitrous oxide ($N_2O$), sulfur dioxide ($SO_2$), methane ($CH_4$), ozone ($O_3$), and fluorinated gases (F-gases) such as hydro-fluorocarbons ($HFC_s$), sulfur hexafluoride ($SF_6$), and chlorofluoro-carbon (CFC) all fall into the category of GHG [2] as shown in Figure 1. On a global scale, four constituents of GHG, namely $CO_2$, $N_2O$, $CH_4$, and F-gases, have seen the greatest increases caused by human activities. The percentage contributions of the individual GHG to total worldwide GHG emissions [3] are shown in Figure 2.

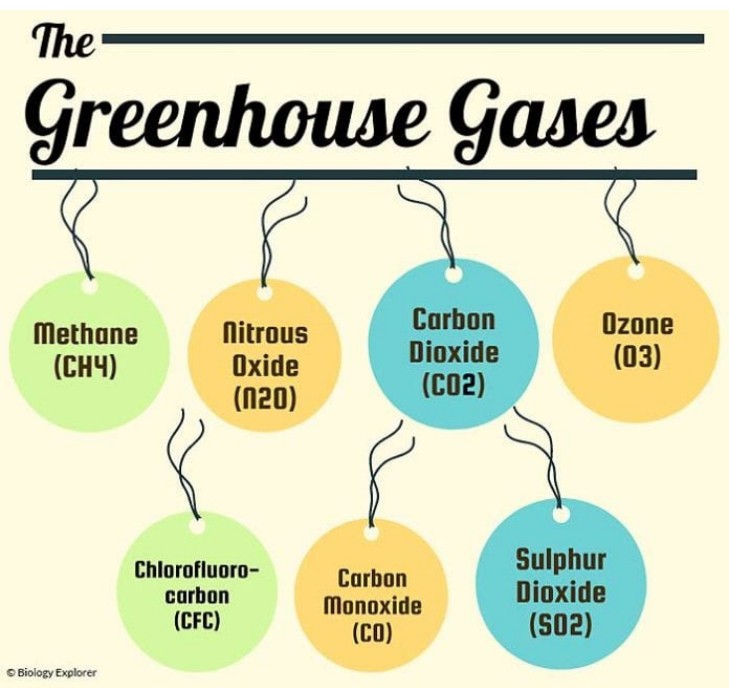

**Figure 1.** Types of greenhouse gases (GHG) that contribute to global warming (Source: [2]).

GHG are emitted as a result of many processes, including the burning of fossil fuels, the clearing of forests for development and industrial use, and the fermentation of sugars by intestinal bacteria. Photosynthesis and photolysis, as well as atmospheric reactions involving oxygen and hydroxyl groups, remove these gases from the environment. Infrared light is absorbed by the residual GHG, which indirectly affects the amount of $O_3$ in the stratosphere, and at high enough concentrations, they contribute to global warming [2]. The release of heat is slowed by the increased temperature of these gases. The melting of polar ice sheets and mountain glaciers directly contributes to increased global temperatures and a commensurate rise in sea levels [4]. In the absence of photosynthesis, low-lying areas would flood, crops and animals would suffer, and human civilization as we know it would come to an end. The most consequential effect occurs when the increased concentration of $CO_2$ in the atmosphere has an impact on world average temperatures. According to Olivier and Peters [5], $CO_2$ is the largest contributor to total worldwide GHG emissions, accounting for around three-quarters—i.e., 74.3% of the emitted GHG was $CO_2$, compared to $CH_4$ (17.3%), $N_2O$ (6.2%), F-gases (2.1%), and other gases (0.1%), as shown in Figure 2.

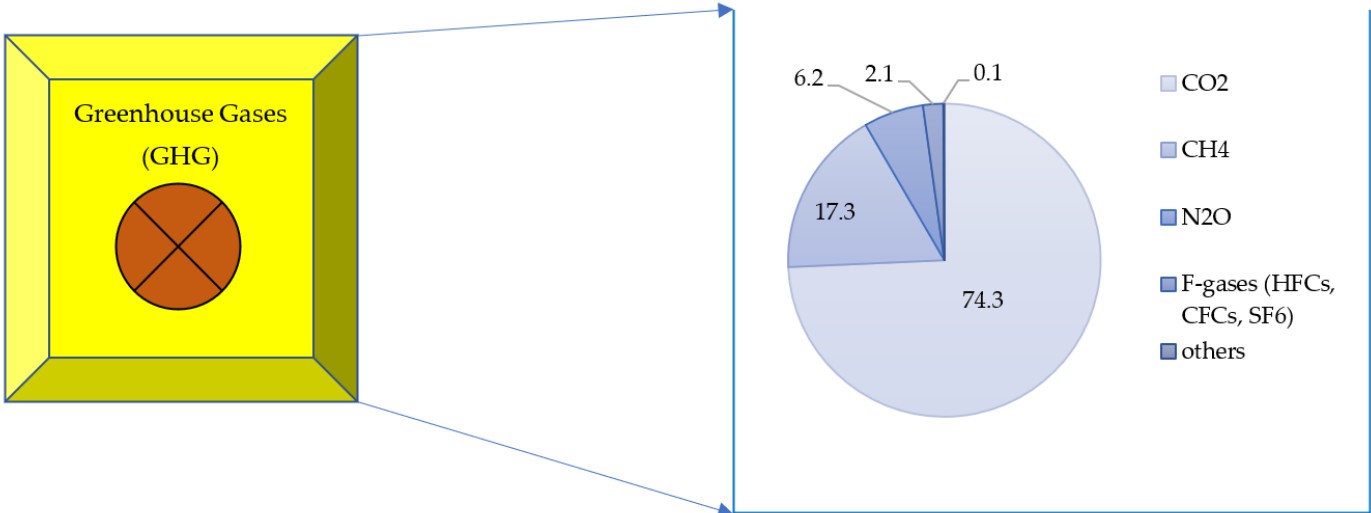

**Figure 2.** Percentage contributions of individual GHG to total worldwide GHG emissions (Source: [3]).

A substantial amount of $CO_2$ has been released into the atmosphere, suggesting that this gas is a primary cause of climate change and global warming. Emissions of $CO_2$ come from a wide variety of human activities [6,7]. These include power plants, commercial and residential buildings, farms and ranches, logging and other land uses, factories, and automobiles. Urbanization and transportation are the largest contributors to $CO_2$ emissions [8] due to the centrality of transportation to the majority of human activities. According to the United States Environmental Protection Agency (EPA) [3], in 2020, 33 percent of all $CO_2$ emissions were expected to come from transportation, 31 percent from power generation, 16 percent from industry, 12 percent from commercial and residential buildings, and 8 percent from the burning of non-fossil fuels. Anxieties have been voiced over the steadily rising volume of $CO_2$ emissions from the transportation sector. The variables, such as the types and amounts of fuel used, that are causing the rising quantity of $CO_2$ emissions, and thus contributing to global warming, need to be better understood immediately.

The rate at which heat is released from the stratosphere is slowed by an increase in the concentration of GHG Increased atmospheric $CO_2$ concentrations have a significant effect on global average temperatures, which is the most serious repercussion. The goal of this paper is to assess the current state of knowledge regarding the methods used to measure $CO_2$ emissions from the transportation sector by developing and implementing a fundamental research method that distinguishes between the different types of measurement and analysis of $CO_2$ emissions used in previous studies and appraises them against current policies concerning green environmental technology that have been developed as part of the effort to achieve zero $CO_2$ emissions by the year 2050 [9].

The environmental impact of mobility has been quantified in previous research; however, the terminology and methodologies used fluctuate wildly. This review focuses primarily on the distinction between $CO_2$ emission measurement techniques and assessment methodologies. It encourages the adoption of accurate approaches for estimating $CO_2$ emissions connected to transportation and raises public awareness of environmental issues. It is hoped that using the best available method will help reduce transportation-related $CO_2$ emissions. The remainder of the review is organized as follows. The methodology of the study is explained in Section 2. The $CO_2$ emissions framework from earlier studies is described in Section 3, while transportation sustainability and development programs are presented in Section 4 with a focus on the key findings that may be of interest to researchers and policymakers in order to ensure sustainable transportation in China. Finally, the investigation is summarized and concluded in Section 5.

## 2. Methodology

This review was performed in accordance with the PRISMA (Preferred Reporting Items for Systematic Reviews and Meta-Analyses) guidelines. A PRISMA flow diagram for a systematic review was successfully developed using a publicly accessible official application called Shinyapp (weblink: https://estech.shinyapps.io/prismaflowdiagram/ accessed on 10 January 2023), as shown in Figure 3. In order to decipher the PRISMA flowchart, it is important to keep in mind the following details:

(1) Data source: Google Scholar, Science Direct, PubMed, Clarivate Analytics Web of Science databases, and e-Journal Portal of Southeast University (SEU) were used with the following title/keywords (no. of strings): GHG emission (446,000); $CO_2$ emissions (2,920,000); $CO_2$ emissions methods (2,340,000); $CO_2$ reduction (5,070,000); GHG transportation (745,000); GHG emissions through remote sensing (RS) and GIS (18,100); GHG policy implications (753,000); GHG mitigation and challenges (365,000); $CO_2$ emissions in Asian countries (179,000); $CO_2$ emission from transportation (751,000); low carbon cities (2,840,000); sustainable development (3,790,000); transportation $CO_2$ emissions monitoring and low carbon city policies (78,800).

(2) Article screening: Records found in the searched databases that were duplicates or irrelevant, that did not have full-text availability, or that were inaccessible or written in languages other than English were excluded and removed from the identified records.

(3) Article inclusion: Only relevant and suitable articles that discussed GHG and $CO_2$ emissions were included.

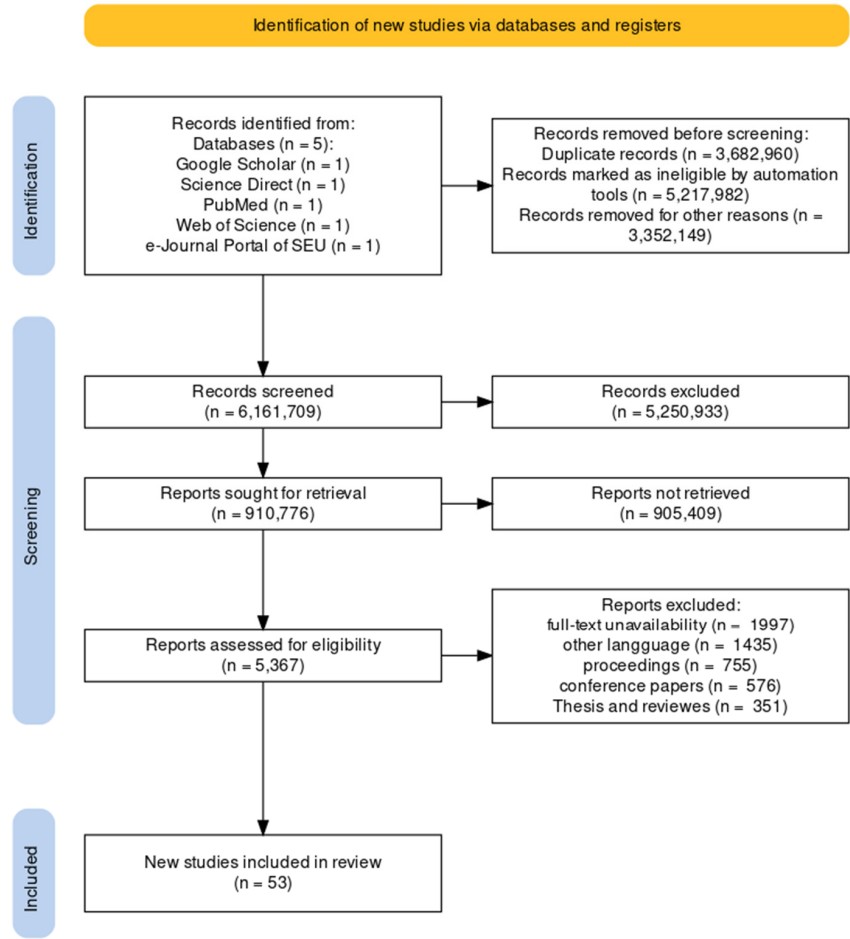

**Figure 3.** PRISMA flow diagram of current study.

As seen in Figure 3, results from searches of Google Scholar, Science Direct, PubMed, and Clarivate Analytics Web of Science for the aforementioned keywords and titles yielded

records from a variety of different sources. In order to reduce the number of records that needed to be sorted through, duplicates were filtered out based on their relevance to the original search keywords used. In the end, 53 research publications discussing $CO_2$ emissions measuring techniques and analysis methods were deemed to be appropriate based on the aforementioned inclusion criteria.

### 3. Framework of $CO_2$ Emissions from Past Studies

This section discusses the transport sector's contribution to GHG emissions, how these emissions are estimated, and how previous research on the topic has been analyzed.

### 3.1. $CO_2$ Emissions from Transport Sector

The growing use of automobile transportation results in higher emissions of GHG, particularly $CO_2$, which has a negative effect on the environment and speeds up the process of global warming and climate change. The $O_3$ depletion is exacerbated by the burning of gasoline in vehicle engines, which releases $CO_2$ and $NO_2$. These engines not only affect the environment by using fossil fuels, but also by emitting harmful pollutants into the air [10]. Transportation modes have a significant impact on air quality [11]. Unless serious measures are taken to address the issue, $CO_2$ emissions from the usage of motor vehicles are projected to increase by 305% by 2050 [12,13], as shown in Figure 4.

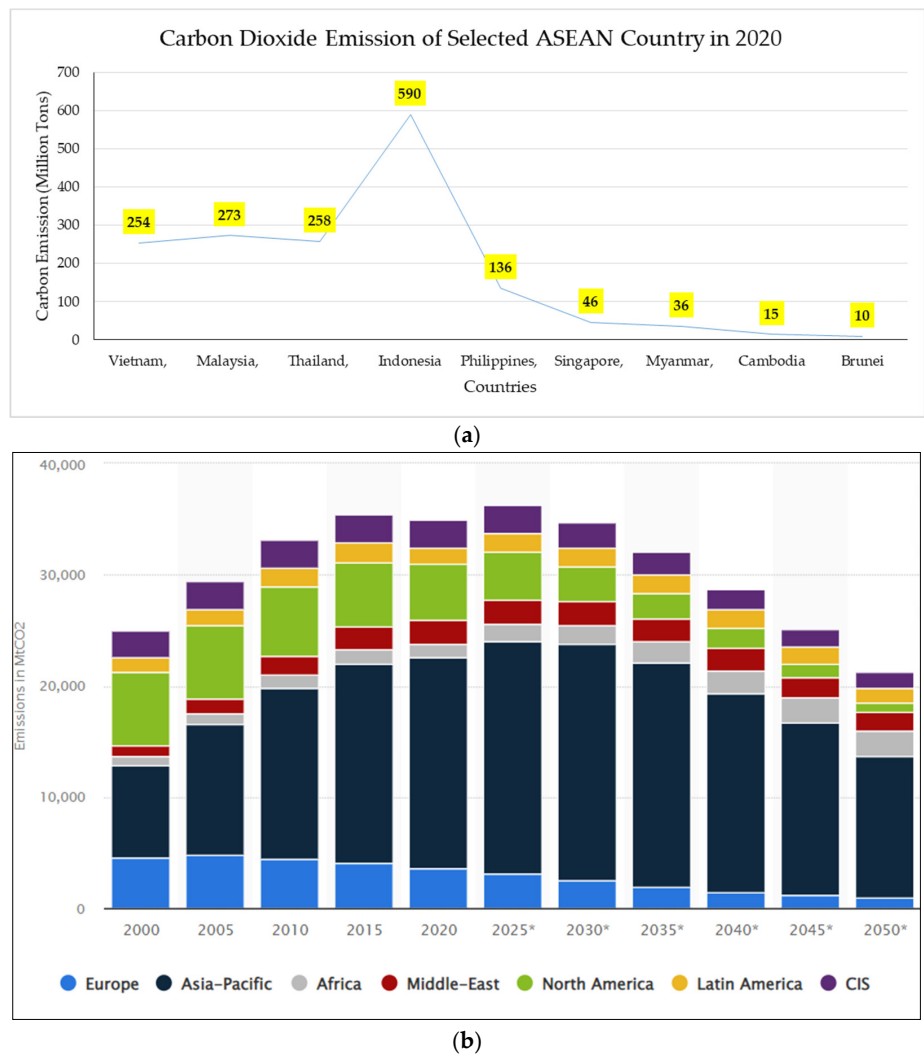

(a)

(b)

**Figure 4.** $CO_2$ emissions: (**a**) ASEAN countries in 2020; (**b**) worldwide from 2000 to 2050; symbol "*" indicated predicted values from 2025 to 2050; (Source: [14]).



According to research by Sukor et al. [15] on transportation-related $CO_2$ emissions, more $CO_2$ is released into the atmosphere when dangerous pollutants from burning fossil fuels react with chemical molecules. The $CO_2$ emissions are influenced by the vehicle's age and the fossil fuel it runs on. Small trucks produce less carbon monoxide than larger vehicles, while vans produce more than both [16]. According to Mickunaitis et al. [17], diesel generates 2.7 kg of $CO_2$ per liter and gasoline generates 2.4 kg per liter. These authors have shown that $CO_2$ emissions are linear with fuel usage, and thus diesel-powered vehicles emit less $CO_2$ than gasoline-powered vehicles. The results of their research show that the size of a vehicle's engine has a direct correlation with its fuel use and $CO_2$ emissions, and that a vehicle's weight also increases its fuel use and hence its $CO_2$ emissions. Every 100 kg added to a vehicle increases its FC (fuel consumption), which in turn raises its $CO_2$ emissions by 6.5% for gasoline-powered and 7.1% for diesel-powered vehicles. In their analysis of the implications of carbon footprint management for road freight transport, Li et al. [18] proposed a method for calculating $CO_2$ emissions that accounted for load and speed, and they suggested further study into how carbon pricing decisions made by multiple organizations might affect the total amount of $CO_2$ emissions produced by different modes of transportation. Paris's attempts to decrease $CO_2$ emissions from transportation faces challenges such as the absence of worldwide legally binding agreements and the significantly higher cost of green solutions, as observed by Santos [19]. Changing the relative costs of sustainable technologies necessitates collaboration between environmental groups and those who set levels of environmental taxes and subsidies. The effect of traffic gridlock on GHG emissions from automobile transportation is currently a hot topic in legal and ethical theory. According to research by Bharadwaj et al. [20], driving in heavy traffic increases both FC and $CO_2$ emissions by 53%. In line with this finding, Zhang et al. [21] discovered that daily peak-hour congestion results in the highest fuel use and $CO_2$ emissions. For this reason, reducing traffic congestion can also reduce $CO_2$ emissions. The differentiation between congested and free-flow phases is crucial for calculating $CO_2$ emissions, assessing exposure and health risk, and organizing transportation. Many countries are banning the use of cars with internal combustion engines in an effort to reduce pollution and improve the health and safety of their populations. These countries include the United Kingdom, Taiwan, and numerous others in Europe. The United Kingdom has pledged to end sales of gasoline and diesel vehicles by 2040 and to achieve emission neutrality across the country by 2050. The government of Taiwan has pledged to end production of internal combustion engine motorbikes and four-wheel-drive cars by the years 2035 and 2040, respectively. In an effort to mitigate climate change, Europe plans to cut its $CO_2$ emissions by 40 percent by 2030 and eliminate them entirely by 2050. Given this action by some governments, rather than enacting more stringent sanctions, governments throughout the world should provide incentives for current car owners to switch to "clean" alternatives. Saighani and Sommer [22] argue that it is possible to minimize transportation-related $CO_2$ emissions through the combined efforts of developing more fuel-efficient automobiles and encouraging safer driving habits. There are a number of ways to do this, including the development of more efficient engines and improved aerodynamics, the manufacture of lighter cars, and the introduction of hybrid electric vehicles. Even with the most optimistic projections, the results of these policies will not be seen for at least two decades. Schroder and Cabral [12] analyzed FC and anticipated $CO_2$ using a 3DRM (3D routing model) and came to the conclusion that eco-friendly strategies can cut FC by 20% and, in turn, reduce emissions. However, due to their greater length, environmentally friendly roads have higher development costs. It is important to recognize that the battle to lower transportation-related $CO_2$ emissions is fraught with controversy, since manufacturers are only prepared to participate in sustainable growth if they can see a return on their investment.

*3.2. CO$_2$ Emission Measurements*

3.2.1. Commonly Used Methods Based on Dataset Type

Measurements of CO$_2$ emissions have been conducted in a variety of ways in previous research. There are two common approaches to estimating CO$_2$ emissions: one uses vehicle characteristics and speed to produce an estimate [23], while the other uses FC and distance traveled [24]. Air-quality monitoring tools can also be used to assess CO$_2$ emissions [25]. To achieve more precise readings, scientists have tried a variety of approaches to calculating CO$_2$ emissions. When it comes to data, time, and cost, each of these techniques has its own set of pros and cons [23].

Under a policy issued by the Asian Development Bank, CO$_2$ emissions are calculated using the total distance driven, average fuel economy, average speed, and kind of vehicle used [26]. The United States backed this policy because it recognizes that different factors—such as vehicle and fuel features, including engine type and technology, air conditioning, fuel qualities and quality, deployment, and the efficacy of maintenance programs—can affect the total amount of carbon dioxide released by vehicles. Transport-related carbon dioxide emissions must be calculated taking account of these factors. The CO$_2$ emissions per mile are calculated by multiplying the total distance traveled by an individual by the appropriate factor, depending on the mode of transportation used. This rather straightforward formula appears in [27–29]. However, while CO$_2$ emission factors may have been established for specific cities, this information is not transferable to studies conducted in other cities [8]. This is due to the fact that different vehicle types, passenger-carrying capacities, and engine sizes all affect how much pollution a given vehicle produces. Further, peak-hour traffic volumes vary considerably between the research locations. Here, *Ex* represents a person's CO$_2$ emissions from their commute, *Sx* represents that person's commute distance, and *C* represents the CO$_2$ emissions factor of the transport mode [30]:

$$Ex = Sx \times C \tag{1}$$

Because CO$_2$ emissions are often lower for shorter travel distances, Wei and Pan [30] stated that the use of new green technology to promote energy efficiency and the design of mixed land use were necessary to protect the environment. Congestion-prone places are particularly well-suited for taking advantage of carpooling's ability to cut down on CO$_2$ emissions by using fewer cars to carry more passengers.

Some scientists have calculated CO$_2$ output from fuel use. When more fuel is used, more CO$_2$ is produced. This approach was used to determine CO$_2$ emissions by Tarulescu et al. [31], who applied a universal coefficient for emissions based on these three variables: first, at 0.249 tons of CO$_2$ equivalent per megawatt-hour (MWh), is the spark-ignition engine; second, at 0.267 tons, is the compression-ignition engine; and third, at 0 tons, is biofuels. They did this by using Equation (2), where *EUsedFuel* is the energy utilized by cars per mile driven, Dr is the route distance in kilometers, Fcaverage is the average FC in liters per kilometer, and Z is the conversion factor for each fuel in kilowatt-hours per liter, with Z = 9.2 for gasoline and Z = 10 for diesel.

$$EUsedFuel = Dr * Fcaverage * Z \tag{2}$$

To calculate CO$_2$ emissions, Aksoy et al. [32] factored in vehicle weight, technical data, and route distance to an existing FC model. It was discovered that the amount of CO$_2$ emissions is proportional to the FC and the distance traveled. The formula developed by Sukor et al. [15] to calculate emissions is shown in Equation (3), where *E* is the amount of CO$_2$ emissions due to transportation (in kilograms), *C* is the amount of gasoline consumed by the vehicle (in liters per one hundred kilometers), and *Fe* is the fuel-based emission factor (in kilograms per liter).

$$E = C * Fe \tag{3}$$

In order to determine how much energy the network as a whole consumes, the measurement takes into account how fast its vehicles are moving on average. According to Grote et al. [33], metropolitan route networks have implemented average speeds for vehicles, which drivers must respect, making this strategy appropriate for urban settings. One of the problems with this approach is that it does not factor in how fast each vehicle is actually going, which is important because vehicles have varied operating characteristics and produce different quantities of $CO_2$. However, it is worth mentioning that when average speed was employed in the computation, identical results were found [34]. Using fuel efficiency as a proxy for distance traveled, Gharineiat and Khalfan [23] determined $CO_2$ emissions according to the formula in Equation (4), where Vs is the mean network speed in kilometers per hour, *Fc* is the FC rate when idling in milliliters per hundred kilometers, *c* is a regression coefficient, and *K* is an adjustment factor that accounts for differences in vehicle attributes across the fleet. The constant term takes into account the influence of drag, inertia, and grade on FC (Kt).

$$S(x) = Fc \times Vs + Kt \tag{4}$$

Vehicle type is also a useful factor for estimating $CO_2$ emissions. The $CO_2$ standard emission factor for trucks is higher than that for light vehicles; however, given that there are far more light automobiles on the road, the overall quantity of $CO_2$ released is higher for light vehicles. The discharge factor for this method was estimated by categorizing vehicles according to their fuel type, with light vehicles using gasoline and heavier vehicles using diesel. The formula for this approach, as used in [23], is displayed in Equation (5). For each pollutant and vehicle type, we can calculate the expected speed-based emission (*Se*) in sin g/km, the city-cycle emission factor (*CCe*) in g/km, and the speed-related function, *T(x)*.

$$S_e = CC_e \times T(x) \tag{5}$$

A summary of the most common techniques employed in the existing research to quantify $CO_2$ emissions is provided in Table 1. Calculations were done taking into account the vehicle's speed, the amount of gasoline used, and the total distance driven. Since the majority of researchers used a consistent methodology across investigations, this review analyzed their findings, taking into consideration that some study characteristics can vary depending on the criteria utilized. The use of air-quality-monitoring equipment in conjunction with the growth of the green transportation industry is a new and encouraging strategy. Several factors, such as VT (vehicle type), FT (fuel type), and TD (travel distance), are included in a standard approach for estimating transportation-related $CO_2$ emissions.

The detection of $CO_2$ in the atmosphere is a burning issue at the moment, and thus researchers are paying more attention to the topic through the use of air-quality-monitoring tools. Air quality was tested by Obanya et al. [25] using handheld instruments that had been calibrated. This method is efficient since it allows for the examination of expansive regions in a short amount of time. Air-quality-monitoring tools have the benefit of providing precise readings of all constituent chemicals of GHG [35]. These include, but are not limited to, methane, ozone, and nitrogen oxides. Once prohibitively expensive, modern air-quality-monitoring tools are now within reach of a wider audience [36].

**Table 1.** $CO_2$ emission measurement methods used in past studies.

| Studies | DTM | FCM | VSM | VTM | AQMM |
|---|---|---|---|---|---|
| Aksoy et al. [32] | Y/A | Y/A | Y/A | Y/A | N/A |
| Asian Development Bank [26] | Y/A | Y/A | Y/A | Y/A | N/A |
| Bhautmage et al. [37] | Y/A | N/A | N/A | Y/A | N/A |
| Chang and Lin [8] | Y/A | Y/A | N/A | N/A | N/A |
| Ding et al. [38] | N/A | Y/A | N/A | N/A | N/A |

**Table 1.** *Cont.*

| Studies | DTM | FCM | VSM | VTM | AQMM |
|---|---|---|---|---|---|
| Faiz et al. [39] | Y/A | Y/A | Y/A | Y/A | N/A |
| Goodchild et al. [40] | Y/A | N/A | N/A | Y/A | N/A |
| Grote et al. [33] | Y/A | Y/A | Y/A | Y/A | N/A |
| Illic et al. [27] | Y/A | Y/A | N/A | N/A | N/A |
| Li et al. [18] | N/A | Y/A | Y/A | N/A | N/A |
| Saighani and Sommer [22] | N/A | Y/A | N/A | N/A | N/A |
| Singleton [41] | Y/A | N/A | N/A | Y/A | N/A |
| Shu et al. [42] | Y/A | N/A | N/A | N/A | N/A |
| Sukor et al. [15] | Y/A | Y/A | N/A | Y/A | N/A |
| Tarulescu et al. [31] | N/A | Y/A | N/A | N/A | N/A |
| Wang et al. [43] | N/A | Y/A | N/A | N/A | N/A |
| Wei and Pan [30] | Y/A | Y/A | N/A | N/A | N/A |
| Clements et al. [36] | N/A | N/A | N/A | N/A | Y/A |
| Yuan-yuan et al. [28] | Yes | Y/A | N/A | N/A | N/A |
| Zhang et al. [21] | N/A | Y/A | N/A | N/A | N/A |
| Obanya et al. [25] | N/A | N/A | N/A | N/A | Y/A |
| Zhuang et al. [44] | N/A | Y/A | N/A | N/A | N/A |
| Rusbintardjo et al. [45] | N/A | N/A | N/A | N/A | Y/A |

Y/A: Yes applied; N/A: Not applied; DTM: distance traveled method; FCM: fuel consumption method; VSM: vehicle speed method; VTM: vehicle type method; AQMM: air-quality-monitoring method.

### 3.2.2. Method Selection and Comparison Based on Available Parameters

To better understand how these techniques are used to determine the types of data used in a study and to implement the systematic analysis of $CO_2$ emissions from transportation, this study compared seven $CO_2$ emission measurement techniques based on seven criteria, namely classification, data-gathering methods, output, cost, time, and accuracy. $CO_2$ emissions can be categorized in a number of ways depending on the method used to calculate them; for example, the trip-distance approach classifies emissions based on how and where they were produced, whereas the FC method groups emissions by speed of travel or vehicle type. In the research by [45] and other sources that rely on air-quality-monitoring equipment, findings are based on data taken during peak hours.

Transport-related $CO_2$ emission measurement techniques vary in the information needed for the analysis. Data on both the distance traveled and the emission factor for the mode of transportation used are needed for the distance traveled method. Data on vehicle speeds is necessary for the vehicle speed approach. In order to use the vehicle type technique, you will need information on the emission factor for your chosen mode of transportation and your average vehicle speed. Information on the gasoline used, the distance traveled, the amount of fuel, and the speed of the vehicle are all essential components of the FC method. Since $CO_2$ emissions can be measured directly by the equipment used in the air-quality-monitoring technique, no data is needed to implement it. A questionnaire survey is required to collect data for the traveling distance method, vehicle speed method, vehicle type method, and FC method. Sensors for measuring air quality give more information than simply the overall amount of $CO_2$ emissions: they also report on the amounts of other pollutants such as N2O and CH4 and man-made compounds such as fluorinated ones. Therefore, there are fewer difficulties involved in returning to the site for follow-up research on previous findings.

The study from [46] was adapted to create a grading system for four types of expenditures (equipment, labor and training, operation and maintenance, and instrument calibration), the findings of which are explained below. The cost of air-quality-monitoring equipment is prohibitive when compared to other methods. The cost of surveys to collect data on travel time, gas consumption, and average speed, as well as the cost of training new drivers, are all included in the second type of expenditure, i.e., labor and training costs. High costs result from the need for more than five people to conduct surveys. Surveys require a large number of workers to conductg them and therefore also take a

long time to bear fruit. The use of air-quality-monitoring sensors results in the lowest labor and training cost because only a small team of workers is required for on-site observation. The third cost indicator, operation and maintenance, is low-priced, with only air-quality-monitoring equipment being required, because this approach is largely utilized for processing operational data and as a tool. It is not necessary to spend money on maintaining air-quality-monitoring sensors because it is assumed that they will always function perfectly. The likelihood of an instrument failing is low because of the high standards to which they are built [9]. Finally, regular equipment calibration is essential for assuring data quality. It is essential that the original data be valid in order to derive a trustworthy result from a data analysis, particularly for compliance purposes or for determining the hazard to a population's health. The TRANSCAT website [47] advises annual $CO_2$ sensor calibration using ambient air (which should consist of 350–450 ppm $CO_2$) and a standard span gas (with a known $CO_2$ concentration). ISO standards for gaseous contaminants call for three types of calibration: (1) initial calibration, during which zero air and calibration gas atmospheres are provided for the first time; (2) operational precision checks, during which the zero and span responses of the instrument are checked for drift on a regular basis; and (3) operational recalibration, during which the analyzer surpasses the instrument performance requirement. As a result, moderate costs are involved with instrument calibration.

This investigation concludes with a comparison of the reliability of the information obtained. Having consistent data throughout and a high degree of accuracy are both crucial [48,49]. Bad data leads to bad conclusions. The responses of road users to a questionnaire survey form the basis of all measurements other than those related to air quality. It is well known that respondents' experience makes questionnaire surveys less reliable and more open to bias. However, in the past 20 years, this approach has been favored by scholars, including the authors of [28,30,37,44]. By using a real-time observation that automatically records the data, it is possible to compute and store data over a longer period and with a bigger storage capacity than is possible with conventional air-quality-equipment assessment. Air-quality-monitoring equipment is extremely reliable, operates constantly with excellent precision, and allows for the repeating of observations, as mentioned by Clements et al. [36]. For this reason, tools designed to measure air quality are preferable to manual approaches.

### 3.2.3. $CO_2$ Emission Estimation Based on GIS/RS and Other Applications

Numerous countries are actively working to lessen their carbon footprints by creating low-carbon urban centers. The decrease in vehicular $CO_2$ emissions is an important part of creating low-carbon cities. Scientists have studied the elements that affect $CO_2$ emissions and produced models for $CO_2$ emissions from transportation in order to apply this solution and bring about future reductions in $CO_2$ emissions. Determining the quantity of emissions and then analyzing the $CO_2$ emitted are the most crucial steps in the lowering of emissions.

$CO_2$ emissions were studied by Ding et al. [38] using a popular decomposition methodology for quantitative analysis. $CO_2$ emission changes were broken down into their component parts using this strategy, allowing researchers to pinpoint the relative importance of each contributing element. Wang et al. [43] developed a decomposition approach by following two steps: (1) incorporating the distinctive features of road freight transportation and (2) considering them in relation to their state's transparent data. The outcomes reported in this study can be considered reliable. The results of the decomposition approach suggested that the following four methods can be used to decrease transportation-related $CO_2$ emissions: (i) regulating the intensity of transportation energy emissions; (ii) enhancing the efficiency with which transportation energy is utilized; (iii) optimizing the transportation structure; and (iv) controlling the demand for transportation.

In order to calculate the whole scope of $CO_2$ emissions, Zhuang et al. [44] employed a mathematical model that factored in both direct and indirect gas emissions throughout the entire production process. Because it takes into account the habits of actual Chinese

people, this technique offers more accuracy than other methods. Tarulescu et al. [31] also proposed the mathematical technique polynomial regression for analyzing $CO_2$ emissions. An increase in fuel use led to a proportional increase in $CO_2$ emissions. Consequently, reducing $CO_2$ emissions can be accomplished by the promotion of bicycle lanes, car fleet renewal, and green urban public transport systems, such as the use of hybrid autos.

The spatial data collected for the purpose of studying emissions can be evaluated and analyzed with the help of GIS. By using a grid squares analysis of $CO_2$ emissions, Byrne and Donnelly [50] were able to visualize and numerically express the results of the model's computations using GIS. To help local decision makers in a novel way, different from conventional GHG inventories, Asdrubali et al. [51] created a GIS-based tool for GHG. Transportation $CO_2$ emissions were estimated using route scale data by Shu et al. [42]. Two authors who also did this were Dalumpines [52] and Chen and Crawford [53]. Land-use shifts were detected with a remote sensing app, and a theme categorization map was generated for use in GIS for further study. Both statistical analysis [54] and geographical analysis are viable options when using a GIS program. A GIS program was shown to be able to efficiently evaluate and handle a wide variety of data formats, which aided in decision making [55].

Five methods for assessing $CO_2$ emissions were proposed by Grote et al. [33]. To calculate the instantaneous emissions of a given mode of transportation, one can use any of the following models: an urban area's average speed-based emissions model; a traffic situation-based model that accounts for congestion; a traffic variable-based model that influences traffic movement capacity in certain cases; a traffic modal-based model that accounts for the types of vehicles people use to get around; a cycle-based model that requires data on the number of stops per kilometer, average speed, and maximum acceleration; and a modal-based model that accounts for the types of fuels people use to get around. Bharadwaj et al. [20] also studied $CO_2$ emissions during rush hour and found that they increased when traffic was heavy compared to when it was light.

Schroder and Cabral [12] employed cutting-edge technologies in a 3D routing model to examine gas mileage and calculate $CO_2$ emissions using GIS software. Using this model, we were able to find the most fuel-efficient routes, compute the total FC as a function of trip time and distance, and then depict these routes in an in-depth 3D profile. A summary of the studies on GHG/$CO_2$ measurement using GIS/RS and other applications are displayed in Table 2.

**Table 2.** Summary of studies on GHG/$CO_2$ measurement using GIS/RS and other applications.

| Author | Studied Region | | Analytical Tool Used | Output |
|---|---|---|---|---|
| | Region | Area | | |
| Asdrubali et al. [51] | Italy | Spoleto | GIS modeling | |
| Bharadwaj et al. [20] | India | Mumbai | VKT, FCM | |
| Byrne and Donnely [50] | Ireland | Greater Dublin Area | GIS modeling | |
| Chen and Crawford [53] | Australia | Melbourne | GIS and RS modeling; Malaysian smart grid | |
| Dalumpines [52] | India | Ahmedabad | GIS and RS modeling | |
| Ding et al. [38] | China | | LMDI method | |
| Grote et al. [33] | United Kingdom | Southampton | traffic variable emissions models | GHG or $CO_2$ emission |
| Idris and Mahmud [56] | Malaysia | Kuala Lumpur | CFA model | |
| Illic et al. [27] | Europe | Serbia | GIS modeling, mathematical sub-models (COPERT IV, CALINE3) | |
| Lorena et al. [57] | Europe | Bucharest, Romania | CFA model | |
| Ma et al. [58] | China | 8 economic zones | ArcGIS and GeoDa software | |
| Schroder and Cabral [12] | Portugal | Lisbon | GIS, 3D-Routing-and DEM model | |
| Shu et al. [42] | USA | State of Louisiana | ArcGIS 9.2, distance-decay principle | |

**Table 2.** *Cont.*

| Author | Studied Region | | Analytical Tool Used | Output |
|---|---|---|---|---|
| | **Region** | **Area** | | |
| Tarulescu et al. [31] | Europe | Ghimbav City | mathematical model, vehicle fleet renewal, RAT Brasov | |
| Wang et al. [43] | China | Beijing | PLSR method | |
| Yazid et al. [54] | Indonesia | Sumatra | CFA model | |
| Yousefi-sahzabi et al. [55] | Japan | Fukuoka | GIS modeling and mapping, prediction dispersion model | |
| Yuan-yuan et al. [28] | China | Xian | correlation analysis and spatial model | |
| Zhang et al. [21] | USA | Michigan | CMEM (MOBILE6.2) | |
| Zhuang et al. [44] | China | Shijiazhuang City | CFA model | |

Here, VKT: vehicle kilometer travelled; FCM: fuel consumption-based method; LMDI: logarithmic mean divisia index decomposition method; CFA: carbon footprint analysis; PLSR: partial least squares regression; CMEM: comprehensive modal emissions model.

*3.3. Summary of Conducted Systematic Literature Review (SLR)*

Five methods have been used in the past to assess $CO_2$ emissions: odometer readings, fuel economy, speed, vehicle type, and air-quality-monitoring devices. A thorough literature analysis concluded that measuring $CO_2$ emissions from vehicles with air-quality-monitoring sensors is the most effective technique due to its short observation duration, real-time data, and capacity to also offer information about other GHG. The goals of the research should be taken into account when deciding on an appropriate measurement method. Similarly, several methods have been used in the past to dissect $CO_2$ discharges. Analysis techniques are selected for specific studies after considering the nature of the data and the goals of the research. Models with the lowest combined FC and yearly mileage are recommended by Gharineiat and Khalfan [23]. Accordingly, identifying the available data is a prerequisite to selecting the best technique for analysis.

This SLR has provided the necessary foundation for gathering and evaluating $CO_2$ data. Twenty research papers were selected for a detailed examination due to their similarities to the analyses performed by prior studies [38,43]. However, some of the investigations have been broadened to utilize multiple aspects, depending on the scope of the studies. In addition, the advantages and disadvantages of the abovementioned five methods (DTM, FCM, VSM, VTM, and AQMM) are summarized below.

The equipment cost ($E_{cost}$) in AQMM is very high. The requirement of labor in AQMM is low and no training is required. The operating and maintenance cost ($OM_{cost}$) of AQMM is low. Alternatively, the other four methods (DTM, FCM, VTM, and VSM) have no $E_{cost}$ or $OM_{cost}$, but their labor requirement is expensive due to the need to distribute the questionnaire for collecting information. In AQMM, real time data is required, while DTM, FCM, VTM, and VSM used data based on questionnaire feedback. Data classified in AQMM was based on peak hours (morning, noon, evening), while in FCM it was based on fuel used, and in DTM, VTM, and VSM mode of transport was used. The data requirements for $CO_2$ emissions calculation using DTM, FCM, VSM, and VTM are travel distance, transport emission factor, type of fuel and transport mode, distance traveled, FC, vehicle speed, and vehicle usage. In contrast, AQMM does not need any type of input as it is based on handheld tools or devices and requires only a few workers to observe $CO_2$ emission. The output obtained from AQMM is very fast, while output in DTM, FCM, VTM, and VSM depends upon the workers' progress in distributing the questionnaire. One key reason to choose AQMM for several researchers [28,30,36,37,44] was to measure all types of GHG, while only $CO_2$ emission can be estimated using DTM, FCM, VTM, and VSM.

**4. Transportation Sustainability and Development Programs**

Several national governments have given the green light to the construction of low-carbon cities (LCC) in an effort to achieve sustainable development that can protect the

world from the long-term effects of global warming and climate change. Using the most up-to-date data and estimations, the Sustainable Development Goals Report 2022 gives a worldwide overview of the implementation of the 2030 Agenda for Sustainable Development. Using in-depth analysis of chosen indicators for each of the 17 Goals, it charts global and regional progress towards achieving the Goals. According to the report, the 2030 Agenda for Sustainable Development and the very existence of mankind are at severe risk as a result of cascading and interconnected problems. The report emphasizes the gravity and scope of the problems we face. All of the Sustainable Development Goals (SDGs) are being negatively impacted by a confluence of problems, including COVID-19, climate change, and wars. The report explains how years of gains in eliminating poverty and hunger, enhancing health and education, and delivering basic amenities have been reversed. It also highlights critical areas where immediate action is required to save the SDGs and bring about substantial development for people and the world by 2030, as seen in Figure 5 [59].

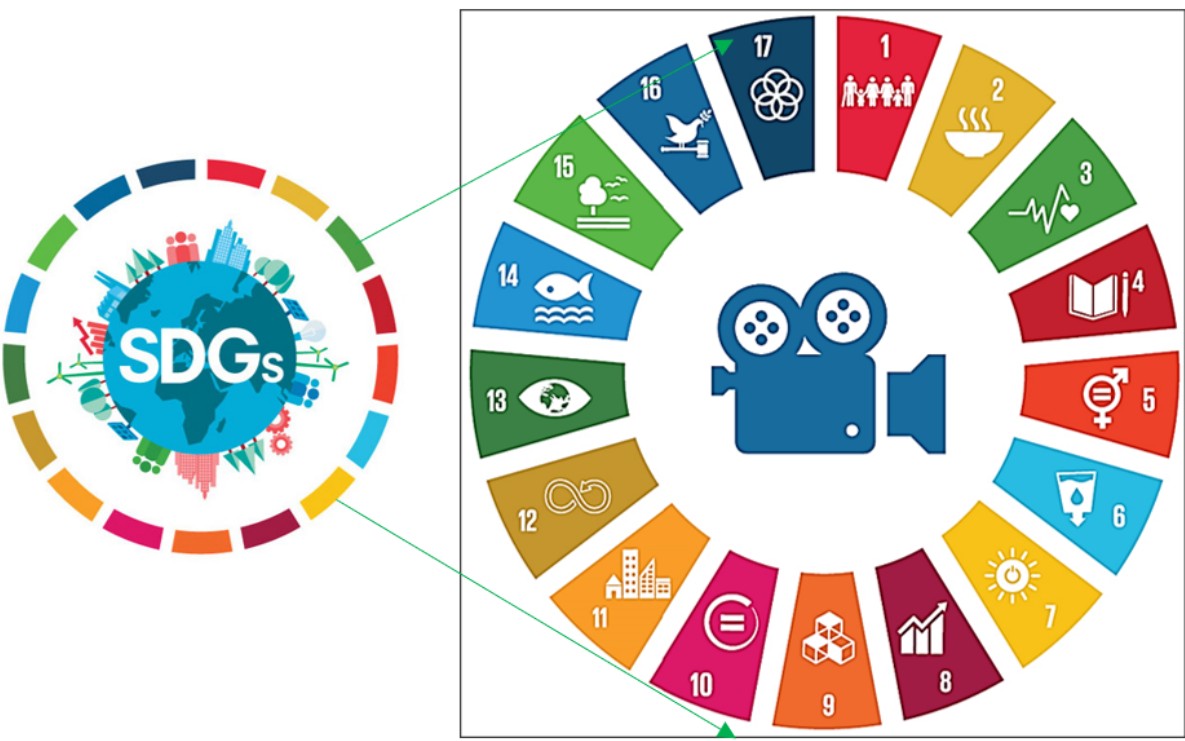

**Figure 5.** 17 Elements of the SDGs. (Source: [60]).

Management, roads, speed, vehicles, road users, and post-crash care are only some of the areas where the SDGs call for improvement [60]. These principles aim to accelerate the growth of human capital so that nations can become developed quickly and sustainably, enhance the welfare of all their residents, and boost the degree of well-being for everyone. As one of the most powerful accelerators for attaining the SDGs, better public transportation can have a significant impact on environmental sustainability. Each of the SDGs can be linked to progress in the transportation sector, including those addressing food security, health, energy, climate change, infrastructure, cities, and densely populated regions. Most urban regions' public transportation systems are unsafe, inefficient, and unsustainable. These things make the gap between the rich and the poor even larger and negatively affect the already vulnerable members of the low-income category. It is critical that we immediately implement a green transportation system that uses renewable energy sources which do not contribute to global warming in the same way that fossil fuels do. The transportation industry is responsible for 25% of global emissions of GHG from energy consumption, and its emission rates are increasing faster than those of any other

industry. Other environmental challenges, such as building public transportation networks that are resilient to environmental disasters and shifting weather patterns, also require immediate attention.

*Road Traffic Mortality Rate (RTMR) in China*

The number of people who died on the roads due to traffic accidents during a given period is referred to as the road traffic mortality rate (RTMR). China is the biggest country by population, and the yearly RTMR in different cities can be found in Figure 6a while the mean RTMR is indicated in Figure 6b. It can be seen that 62,763 people lost their lives in road accidents in China in the year 2019. Every year since 2016, almost 63,000 people in China have died in traffic-related incidents. The number of people killed in traffic accidents in China varies widely by location. As of 2012, the greatest rates of traffic deaths were recorded in the provinces of Guangdong and Hubei, as shown in Figure 6a. These provinces are both on China's eastern coast, which had the highest rates of road fatalities in 2020. In contrast, reports of deaths in central and western China were quite low. Lower traffic volume and the lack of dense metropolitan centers might be to blame for this discrepancy. In 2019, there were 62,763 fatalities and 256,101 injuries due to road accidents in China, both considerable increases over 2016. The fact remains, however, that road accidents are now a major source of mortality in China. A lack of road safety awareness among Chinese drivers and uncontrolled road construction may be key causes. With more people using the roads and more automobiles on the road, China's road infrastructure and driving habits have not kept pace. In 2003, just 24 million automobiles were registered in China, but that figure had increased to 253.76 million by 2019. Almost 25.5 million automobiles were registered in China in 2019 alone.

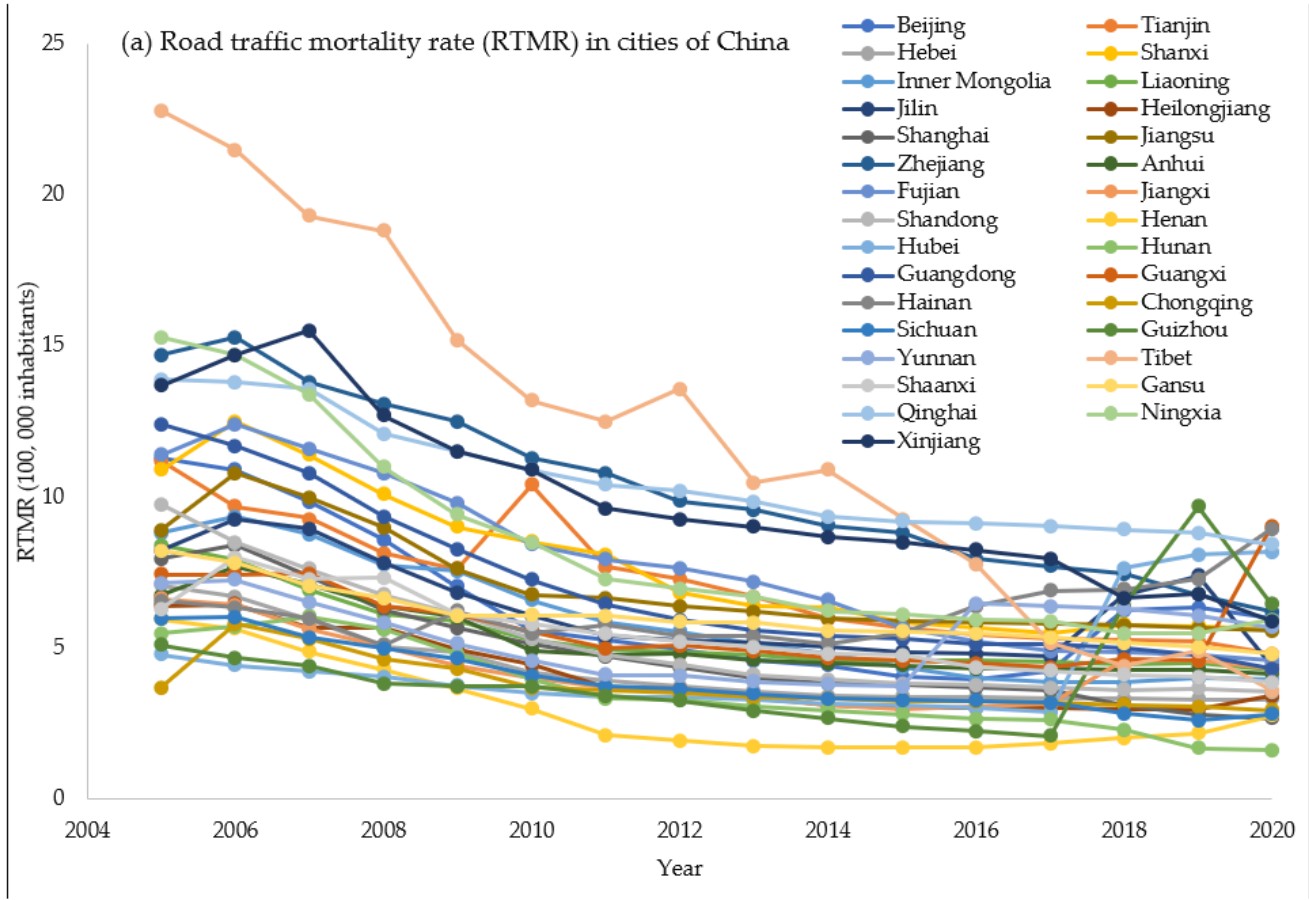

**Figure 6.** *Cont.*

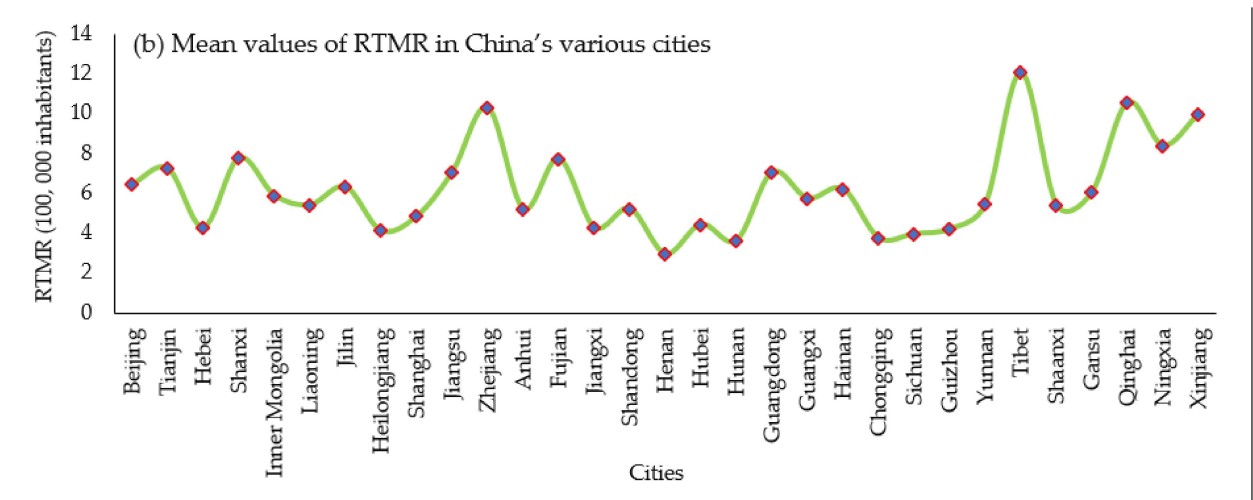

**Figure 6.** Road traffic mortality rate (RTMR) in China's cities: (**a**) yearly values; (**b**) mean values.

The number of people killed in traffic accidents in China has risen, the WHO reports [61,62]. In addition to lowering the frequency of muggings and robberies on public transportation vehicles and at stations, there is a pressing need to develop a system that provides enhanced safety for female passengers from sexual assault and abuse [63–65]. A mechanism that makes it easy for both drivers and passengers to recognize and report human trafficking is also necessary to improve sustainability and development programs in the transportation sector [66,67].

The first-ever NEV requirement, or Parallel Management Regulation for Corporate Average FC and New Energy Vehicle (NEV) Credits, was enacted in China on 1 April 2018 for light-duty cars. The regulation was approved in September 2017. Battery electric cars (BEVs), plug-in hybrid electric vehicles (PHEVs), and fuel cell vehicles (FCVs) are all classified as NEVs in China. The goal of the regulation is to facilitate the manufacturing of new energy vehicles and to give states more leeway in meeting the nation's current FC standard through the use of a credit system based on technological advancements. The rule draws inspiration from the California Zero Emissions Vehicle (ZEV) program and expands on it by including a parallel management component that links manufacturers' corporate average fuel consumption (CAFC) performance with their NEV credit performance. The second phase of the policy started on 1 January 2021. The overall framework is the same as in the first phase, but the standards are tightened to boost China's economic competitiveness and deliver significant social repercussions in terms of accessibility and inclusion, while lowering the transportation sector's negative environmental effect [63]. The transportation industry is one of the primary targets of this policy's ZEV embracement.

Each sale of an NEV results in credits toward meeting the NEV mandate's requirements; the amount of credits earned is dependent on factors such the vehicle's electric range and energy economy and the rated power of its fuel cell systems. As a result, the ultimate fleet mix will determine the final NEV market share reached in China as a result of the effect of the credit objectives. To comply with the first two years of the ZEV requirement (2019 and 2020), automakers were required to meet annual NEV credit percentage objectives of 10% in 2019 and 12% in 2020. In the second stage (2021–2023), the percentage goals are 14% in 2021, 16% in 2022, and 18% in 2023. This will secure the well-being of the population by providing a reliable mode of transportation that promotes economic growth. In conclusion, a sustainable transportation system must play a pivotal role in the plan for long-term economic growth. Because transportation infrastructure is designed to last for decades, current policies made by local and national governments will have long-term consequences for urban expansion and, by extension, on the environment.

The start of a period of exceptional growth for China's transportation sector dates back to the 1990s. The total amount of ton-kilometers converted by all modes of transportation

increased from 29,153 in 1990 to 119,554 in 2008. Increases in the quality of transportation led to a sharp increase in carbon emissions. Since the year 2000, China's transportation sector's carbon emissions have seen exponential rise. Growth in carbon emissions has also been affected to varying degrees by the transportation sector's level of development, energy efficiency, and structure. Over the past two decades, we have seen a steady uptick in transportation's energy efficiency, which has helped slow the rise of carbon emissions. However, transportation's energy efficiency was unable to slow the rise of carbon emissions because of the increasing importance of motorways. After the year 2000, the effect of energy efficiency as a brake on carbon emissions began to weaken as the phase-out of steam locomotives on the railroads progressed. China must take immediate action to reduce $CO_2$ emissions from its transportation sector. Several nations and cities have succeeded in cutting their $CO_2$ emissions from transportation—for example Mexico City (45), China (64), and Shanghai (31). Establishing green zones in which plants with a high ability to absorb $CO_2$ are planted are one of the ways in which cities are attempting to reduce their carbon footprints (Melbourne [53]). Depending on their economic viability and China's present growth strategy, the steps implemented by other nations will be able to be replicated by China in its attempts to successfully cut $CO_2$ emissions from transportation. Government and non-government groups working together on this might potentially be an option.

## 5. Conclusions and Future Work

This article reviews various carbon dioxide ($CO_2$) emissions measurement and analysis methods employed by various researchers in past studies. Despite its crucial role in reducing greenhouse gas (GHG) emissions, $CO_2$ is the major constituent among all GHG gases. The nomenclature and methods used in the calculation of $CO_2$ emissions vary and depend upon the availability of datasets. Unfortunately, the varied methodologies for computing $CO_2$ emissions from transportation have led to misunderstandings and problematic implementations. This research presents five methods, namely DTM, FCM, VSM, VTM, and AQMM, used by researchers to calculate $CO_2$ emissions. Method selection and comparison based on available parameters have been explicitly explained in the current study. In addition, the advantages and disadvantages of the aforementioned methods have also been discussed to facilitate choosing the best one according to location and data availability. It was determined that AQMM has become imperative for the calculation of $CO_2$ emission compared to other methods. Measurement using AQMM came out on top because of its rapid response time and real-time data, as well as the fact that it provides information on other greenhouse gases. By identifying the origins of these emissions, our review has made a substantial contribution to addressing the issue. This work raises environmental awareness and supports accurate transportation-related CO2 emission calculations. However, different approaches have different strengths and weaknesses, and selecting the right one for a study relies on its objectives. Choosing the best technique should reduce transportation-related $CO_2$ emissions.

China is the biggest country by population, and the yearly RTMR in different cities has reduced but is still high in Tibet and Zhenjiang cities. China should also revise its transportation law to prohibit the use of low-occupancy vehicles during rush hours, among other changes; create a database for sustainable transport management to improve analysis and decision making; teach students in elementary and secondary schools, as well as the general public, on why eco-friendly modes of transportation are so crucial; and compel all businesses to conduct $CO_2$ assessments. In the battle to lower $CO_2$ levels, these actions are crucial. By way of conclusion, the transportation sector can succeed in enhancing access and connection, lessening traffic congestion, cutting down on fuel costs, enhancing the environment, and encouraging a healthy way of life. It is important to note that the world intends to reach its target of zero $CO_2$ emissions by 2050. This study summarizes the objectives of the Sustainable Development Goals and includes a global policy for sustainable development. Its findings can also be used to inform the design of environmentally friendly

transportation systems. Thus, it is anticipated that this assessment will point the way toward future opportunities to reexamine established ideas and introduce novel ones.

**Author Contributions:** Conceptualization, M.Z.; formal analysis, M.Z.; methodology, M.Z.; writing—original draft, M.Z.; supervision, S.C.; review and editing, Y.M. and X.H. All authors have read and agreed to the published version of the manuscript.

**Funding:** This research was supported by philosophy and social sciences research projects of universities in Jiangsu, China (No. 2020SJZDA133) and research projects of the Humanities and Social Science Foundation of the Ministry of Education of China (No. 20YJCZH121).

**Institutional Review Board Statement:** Not applicable.

**Informed Consent Statement:** Not applicable.

**Data Availability Statement:** Data available on reasonable request.

**Acknowledgments:** We are thankful to the editor and anonymous reviewers for their helpful comments to improve this manuscript. Special thanks to Shuyan Chen for her supervision and guidance during the whole work.

**Conflicts of Interest:** The authors declare no conflict of interest.

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
