# Peer review of "A Systematic Review on Carbon Dioxide (CO2) Emission Measurement Methods under PRISMA Guidelines: Transportation Sustainability and Development Programs"

_sustainability, doi:10.3390/su15064817_

Round 1

Reviewer 1 Report

Nice presentation

Author Response

The manuscript is convincing and has great importance on Sustainability of transport medium by accurately identifying the CO2 emission from the vehicles. It is well constructed. However, this manuscript can be improved based on the following comments

We are thankful to the reviewer for valuable time to review and suggest improvements on current study to make it publishable. We modified our paper as per suggestions. 

In abstract section, line 18-19; the conclusion seems to be not ideal approach. I would like to suggest making different conclusion.

We revised the lines.

Same this applies for line 96-102. In this line making what you have described in the manuscript by making numbers, sections and others is not consistent writing. Make it different languages what you have described in those sections, figures etc.

We revised the lines.

In keywords, first4 keywords are similar. Normally a good scientific paper ask for 5 key words.

Thank you for your suggestions. We added 5 relevant keywords.

Line 24, populace?

Populace mean citizen, replaced. 

In figure 2; only mentioned gases are not only greenhouse gases, you do not have reliability approach for sure. You do not have others gases data (You make list % as others)

Carbon dioxide (CO2), nitrous oxide (N2O), sulphur dioxide (SO2), methane (CH4), ozone (O3) and fluorinated gases (F-gases) such as hydro-fluorocarbons (HFCs), sulfur hexafluoride (SF6) and chlorofluoro-carbon (CFC), all fall under the category of GHG [2] as shown in Figure 1. At global scale, four constituents of GHG namely, CO2, N2O, CH4, and F-gases have the greatest impact caused by human activities. The global share of GHG emissions contribution with respect to others gases data (%) is added in Figure 2.

In figure 2, what is ‘T’?

In figure 2, w.r.t indicates "with respect to"

In figure 4 b) 2050 is written. Is it predicted? Nothing explained in legend.

Yes, it is projected and symbol added in figure 4 (b) for easily understanding. The explanation is presented in line 143-144 as "Unless serious measures are done to address the issue, CO2 emissions from the usage of motor vehicles are projected to increase by 305% by 2050".

There are two figure 5 in the manuscript.

Corrected

In second figure 5, data is not clearly seen. Make high resolution images. And indicate a) and b) and then describe them what they represent for.

We replaced figure 5 with high resolution.

Support the conclusions with your specific objective.

We added specific objective to support our study conclusion. We revised our conclusion section.   

Reviewer 2 Report

This manuscript reports on the status and measurement of CO2 under PRISMA guidelines. The topic is imperative and fits the scope of this journal. I only have a few minor reservations about this review.

1.      In Section 2, as mentioned by the author, there are numerous duplicates and unrelated records in the database. How did the author define irrelevance and how did they remove these records, artificially? The author needs to clarify.

2.      There may be a formatting error in line 375 of the article, I suggest the author check it.

3.      Section 3.3, what is the meaning of ‘SLR’? The definition of SLR cannot be found in the manuscript. The author should check that the abbreviations appearing for the first time in the paper have exact definitions.

4.      The article provides a detailed review of CO2 measurement methods, but lacks sufficient discussion of the advantages and disadvantages of different approaches.

Author Response

This manuscript reports on the status and measurement of CO2 under PRISMA guidelines. The topic is imperative and fits the scope of this journal. I only have a few minor reservations about this review.

We are thankful to you for acknolwedge our work. We have reivsed our paper as per your suggestions. 

In Section 2, as mentioned by the author, there are numerous duplicates and unrelated records in the database. How did the author define irrelevance and how did they remove these records, artificially? The author needs to clarify.

We used freeware Sysrev (A web-based platform) to remove duplicates and unrelated records and Shinyapp was used to develop PRISMA flow diagram. It facilitates a range of document-review types, from pure data curation projects to systematic reviews. Sysrev Label system enables flexibility in how users screen, tag, annotate, & sort documents and extract data. Articles can be uploaded via .XML, .RIS, and .PDF file types or via Sysrev native search engines. Sysrev BASIC users also have access to machine learning models. In addition, Records found in the searched databases that did not have full-text availability, or were inaccessible or written in languages other than English were excluded and removed. Only relevant and suitable articles that discussed GHG and CO2 emissions method and techniques by reading the abstract and content of article were included. For more information, following are the references:

  1. https://sysrev.com/
  2. https://www.frontiersin.org/articles/10.3389/frai.2021.685298/full

There may be a formatting error in line 375 of the article, I suggest the author check it.

We revised and corrected.

Section 3.3, what is the meaning of ‘SLR’? The definition of SLR cannot be found in the manuscript.

Corrected and updated in whole manuscript

 The author should check that the abbreviations appearing for the first time in the paper have exact definitions.

Corrected and updated in whole manuscript

The article provides a detailed review of CO2 measurement methods, but lacks sufficient discussion of the advantages and disadvantages of different approaches.

Discussion of the advantages and disadvantages of five methods (DTM, FCM, VSM, VTM and AQMM) are added under subsection "3.3. Summary of conducted systematic literature review (SLR)".

Reviewer 3 Report

Good work.

Good luck!

Reviewer 4 Report

A Systematic Review on Carbon Dioxide (CO2) Emission Measurement Methods under PRISMA Guidelines:  Sustainability and Development Programs from Transportation by

Muhammad Zubair was reviewed and Specific comments are given below.

A revision is recommended to this submission.

·       Attempt seems to be good.

·       What is EQA? All the abbreviations should be expanded first time, it will be helpful for the readers for easy understanding.

·       For the calculation of CO2 emissions various methods was tried. It is appreciable all the parameters were taken into account

·       Figure 5 should be explained clearly

·       All the figures must be serially numbered. Figure in page 14 needs more clarity

·       Care should be taken to avoid typographical error especially subscripts

·       The knowledge gap should be clearly stated.

·       References should be arranged as per the journal format.

These corrections would improve the content of the manuscript and significantly increase the citations

Author Response

A revision is recommended to this submission.

We revised as per your valuable suggestion.

Attempt seems to be good.

Thank you for recommendation.

What is EQA? All the abbreviations should be expanded first time, it will be helpful for the readers for easy understanding.

Thank you for improvement. We carefully corrected abbreviations and updated in whole manuscript.

For the calculation of COemissions various methods was tried. It is appreciable all the parameters were taken into account

Thank you for acknowledged.

Figure 5 should be explained clearly

We explained figure 5

All the figures must be serially numbered. Figure in page 14 needs more clarity

Corrected

Care should be taken to avoid typographical error especially subscripts

We carefully revised whole manuscript to omit typographical error especially subscripts.

The knowledge gap should be clearly stated.

We updated introduction section with the knowledge gap.

References should be arranged as per the journal format.

Revised and Corrected

These corrections would improve the content of the manuscript and significantly increase the citations

We modified our paper as per your suggestion. Thank you for valuable time to review and suggest Improvements to make it interesting for readers.